# Determinants of vaccine hesitancy in Switzerland: study protocol of a mixed-methods national research programme

Michael J. Deml [1,2] , Kristen Jafflin,[1,2] Sonja Merten,[1,2] Benedikt Huber,[3] Andrea Buhl,[1,2] Eleonora Frau,[2,4] Valérie Mettraux,[2,4] Joanna Sonderegger,[2,4] Paulina Kliem,[2,4] Rachele Cattalani,[2,4] Daniel Krüerke,[5] Constanze Pfeiffer,[1,2] Claudine Burton-Jeangros,[6] Philip E. Tarr[2,4]

MJD, KJ and SM contributed equally.

For 'Presented at statement' see end of article.

For numbered affiliations see end of article.

**Correspondence to**
Dr Philip E. Tarr;
philip.tarr@unibas.ch

## ABSTRACT

**Introduction** Vaccine hesitancy is a complex public health issue referring to concerns about the safety, efficacy or need for vaccination. Relatively little is known about vaccine hesitancy in Switzerland. This ongoing study (2017–2021) focuses on biomedical and complementary and alternative medicine (CAM) providers and their patients since healthcare professionals play important roles in vaccination decision-making. This national research programme seeks to assess the sociocultural determinants of vaccine hesitancy regarding childhood and human papillomavirus vaccines in Switzerland. We aim to provide a detailed characterisation of vaccine hesitancy, including CAM and biomedical perspectives, patient–provider interactions, and sociocultural factors, to establish the mediating effects of vaccine hesitancy on underimmunisation, and to design an intervention to improve vaccination communication and counselling among physicians, parents and adolescents.

**Methods and analysis** Our transdisciplinary team employs a sequential exploratory mixed-methods study design. We have established a network of more than 150 medical providers across Switzerland, including more than 40 CAM practitioners. For the qualitative component, we conduct interviews with parents, youth, and biomedical and CAM providers and observations of vaccination consultations and school vaccination information sessions. For the quantitative component, a sample of 1350 parents of young children and 722 young adults (15–26 years) and their medical providers respond to questionnaires. We measure vaccine hesitancy with the Parent Attitudes about Childhood Vaccines 15-item survey and review vaccination certificates to assess vaccination status. We administer additional questions based on findings from qualitative research, addressing communication with medical providers, vaccine information sources and perceptions of risk control vis-à-vis vaccine-preventable diseases. The questionnaires capture sociodemographics, political views, religion and spirituality, and moral foundations.

**Ethics and dissemination** The study was approved by the local ethics committee. The results will be published in peer-reviewed journals and disseminated to healthcare professionals, researchers and the public via conferences and public presentations.

### Strengths and limitations of this study

► A novelty of this research includes its large sample of complementary and alternative medicine (CAM) and biomedical providers, with consideration given to their vaccination perspectives and interactions with their patients.

► The study emphasis on CAM providers is important since previous studies have shown an association between CAM use and vaccine hesitancy as well as high patient demand for CAM in Switzerland.

► The sequential exploratory mixed-method study design and transdisciplinary nature of this study provide further insights into the relationships between CAM and biomedical providers and vaccine hesitancy and uptake, with the qualitative methods and results having informed the design of the quantitative questionnaire.

► Although the methodological approach to recruit patients via medical providers is advantageous because it allows for the examination of effects of provider characteristics on patient vaccination beliefs and behaviours, it limits our results because populations that do not regularly see medical providers are under-represented in our data.

► Furthermore, the data do not provide nationally representative results.

## INTRODUCTION

Vaccines are ranked among the greatest public health achievements.[1] Extensive evidence documents the efficacy, minimal side effects and cost-effectiveness of vaccines recommended by numerous national immunisation programmes, including that of the Swiss Federal Office of Public Health (FOPH).[2 3] Despite public health successes of vaccines, a large number of people in Western countries express concerns about vaccine safety, efficacy or need, a behaviour now referred to as *vaccine hesitancy*.[4–6]

Vaccine hesitancy has gained increasing attention from the global public health

community. In August 2015, the WHO referred to vaccine hesitancy as a 'growing challenge for immunization programs',[7–9] responding to a 2014 report[10] from the WHO Strategic Advisory Group of Experts on Immunisation. In early 2019, WHO listed vaccine hesitancy among the top 10 threats to global health, citing a global need to address rising rates of measles and to eliminate cervical cancer by ramping up efforts to increase coverage with the human papillomavirus (HPV) vaccine.[11]

While the importance of vaccine hesitancy is widely acknowledged, its determinants are incompletely understood and merit further investigation.[4] Researchers having conducted a systematic review on determinants of vaccine hesitancy from a global perspective found that there was 'no universal algorithm' (p2155) and recommended context-specific research.[4] Other researchers have argued in favour of cross-disciplinary research, stating 'Vaccine hesitancy and acceptance are complex, multi-faceted issues. Our understanding of them must be complex and multifaceted as well' (p279).[12]

When vaccine hesitancy leads to *underimmunisation*, which we define as delaying or omitting some or all officially recommended vaccines, this provides an example of the underuse of a medical intervention considered to be safe and effective. Underimmunisation is a public health concern because it can reduce population protection, also referred to as *herd immunity*, towards certain potentially dangerous infectious agents and increases the risk of disease outbreaks.

Studies suggest that personal, social[12 13] and local cultural[6] networks are important determinants of vaccination decisions, in particular among those who undervaccinate. Furthermore, people's sources of information on health and vaccination-related information can play important roles in vaccination choices, with recent research pointing to the importance of 'information overload', 'misinformation' and higher levels of indecision among parents, particularly given the wide availability of (mis)information on the internet.[14–18]

Additionally, parent and provider perspectives that differ from official public health recommendations need to be considered in order to understand vaccine hesitancy.[12] These may include intuitive rather than analytical cognitive styles, reluctance to consider the evidence suggesting vaccine safety/efficacy, and lowered willingness to trust information delivered by public health authorities, as shown in an adherence to complementary and alternative medicine (CAM)-oriented health values and preference of CAM over biomedicine.[19] Studies also suggest that underimmunisation and usage of CAM are determined by similar factors, such as alternative concepts of body, immunity, risk perception, antiauthoritarian orientation, and distrust of biomedical establishment and the pharmaceutical industry.[6 19] Some of these associations may in part be due to confounding factors, such as higher income and education and distrust of medical systems.[20] Other sociomedical trends may also play a role in heightened patient participation in decision-making, such as

postmodern medicine,[21] characterised by risk culture,[22] healthism[23] and patients activating more agency in healthcare decisions.[24] Finally, research findings suggest limitations to the efficacy of the evidence-based strategies that have generally been employed by health authorities over the past decades to increase vaccination rates.[19]

A key determinant of patient vaccination decision-making seems to be the attitudes, behaviour and knowledge of their medical provider.[25–27] Important factors include the amount of time providers spend discussing vaccinations, their communication styles and the information available to providers.[28–31] As an example of medical providers' influence on vaccination decisions, one study in the USA found that primary care providers with many parents who had vaccine exemptions for their children were more likely to have concerns about vaccine safety and were less likely to perceive individual and community benefits of vaccinations than primary care providers treating fewer children with vaccine exemptions.[25] A qualitative study into vaccination decision-making in the USA highlighted the importance of trust between healthcare professionals and parents.[32] Finally, the way that providers broach vaccination during consultations can have a determinative role in vaccination uptake, with one study recommending further investigation of participatory (ie, 'What do you think about doing shots today?') versus presumptive (ie, 'Today, we are going to do some shots') approaches during patient–provider interactions.[26]

### The Swiss context

In Switzerland, childhood vaccinations are usually given by private practitioners, predominantly paediatricians and general internists. There is no national mandate for vaccination in non-epidemic settings in Switzerland and no national vaccination registry. Despite this, vaccination coverage in Switzerland is high overall.[33] The FOPH makes vaccination recommendations and distinguishes between basic and and complementary vaccinations. (*Basic* vaccines are defined as 'essential to individual and public health, and offer a level of protection that is indispensable to people's well-being (eg, diphtheria, tetanus, pertussis, polio, MMR (mumps, measles, rubella), HBV (, HBV (hepatitis B), HPV (human papillomavirus)' (pA51). *Complementary vaccines* are defined as being able to 'maximize individual protection and are meant for individuals seeking protection from well-defined risks (eg, conjugate pneumococcal vaccine and conjugate meningococcus C vaccine)' (pA51).[34]) Both basic and complementary vaccines are reimbursed by basic mandatory health insurance when the official vaccination schedule is respected.

Given the absence of a national vaccination registry, the FOPH measures Swiss vaccination coverage at regular intervals.[35] Vaccination programmes and their implementation vary between the 26 cantons.[34] For example, children from the French-speaking and Italian-speaking cantons have on average higher rates of measles vaccination coverage than in German-speaking cantons.[36]

Cantonal coverage levels for HPV vaccination range from 79% (Valais) to 19% (Appenzell Innerrhoden) for two doses of HPV vaccine. HPV coverage estimates only include 16-year-old girls and estimates for boys are not yet available.[37] The most common reasons for women not being vaccinated against HPV, according to a 2014 survey, included being too old, lack of information, fear of side effects, being against vaccination in general or against HPV vaccination only, HPV vaccination felt to be unnecessary, and simple logistical issues,[38] with 7% and 6% of women not vaccinated against HPV reporting that the physician or friends/family recommended against HPV vaccine, respectively. Further research has shown that living in cantons with school-based vaccination increases uptake of HPV vaccination.[39 40] However, spatial variation modelling has shown that variables such as political opinion, religion and community opinion might be additional determinative factors in the presence of school-based vaccination programmes.[40] This suggests further research is needed to better understand vaccine hesitancy and school-based vaccination programmes in Switzerland.

Vaccine hesitancy and underimmunisation seem to be specific to certain population subgroups.[34 41 42] For example, in Switzerland, measles cases, small epidemics[43] and underimmunisation cluster around children attending anthroposophic (ie, Rudolf Steiner, Waldorf) schools and around certain providers of CAM.[41] One cross-sectional survey (n=1007) was administered for paediatric patients in an urban paediatric emergency department in German-speaking Switzerland. Researchers found that those who did not fully accept basic vaccinations were more frequently CAM users than non-users, which researchers point to as reflecting parent wishes rather than physicians' recommendations.[44] This relationship merits further investigation and discussion.

Studies in Switzerland show high rates of CAM use and favourable opinions, with 25% of Swiss survey respondents aged 15 and older of the 2007 Swiss Health Survey stating that they had used CAM in the previous year.[45 46] Other researchers assessed surveys conducted among the general population, doctors, hospitalised patients and obstetric institutions and found that approximately 50% of the Swiss population had used CAM and that 85% of the population would like the cost of CAM to be covered by basic health insurance.[47] CAM providers in Switzerland are often physicians trained in conventional medicine who have then obtained additional CAM training,[48] and basic mandatory health insurance reimburses four CAM services when provided by licensed physicians: (1) anthroposophic medicine, (2) traditional Chinese medicine and acupuncture, (3) homeopathy, and (4) phytotherapy.[49]

### Rationale for the research project

The scientific literature points to existing research gaps, such as a lack of detailed information on vaccine hesitancy, patient–provider communication in Switzerland, vaccination information sources and relations to the high prevalence of CAM use in Switzerland. This highlights the interest in an exploratory mixed-method methods study investigating the vaccination perspectives and practices of patients and both CAM and biomedical providers. Our research focuses specifically on childhood vaccinations recommended in Switzerland[50] and on the vaccination against HPV, which is included in the Swiss FOPH recommendations for all adolescents 11–14 years of age. However, vaccination against HPV is recommended as a routine vaccination for females and as a complementary vaccination for males.[50]

### Objectives and aims

► To explore vaccine hesitancy and provide a detailed characterisation of its health system, patient–provider interaction, health communication, information sources, decision-making process, and demographic, geographical and sociocultural correlates in Switzerland.

► To assess the sociocultural and health system determinants of vaccine hesitancy and underimmunisation with childhood and HPV vaccines in Switzerland. This objective additionally aims at establishing the mediating effects of vaccine hesitancy on underimmunisation.

► To use the knowledge gained as the necessary background to design and submit a pilot intervention and to design tailored interventions to address vaccine hesitancy in Switzerland.

## METHODS AND DESIGN

Our research takes place in the setting of an ongoing Swiss national research programme (National Research Programme 74). The study protocol underwent successful peer review. For further information, refer to http://www.nfp74.ch/en/projects/out-patient-care/project-tarr. It represents a transdisciplinary[51] collaboration that views vaccine hesitancy as a complex, multifaceted phenomenon. Rosenfield[51] explains how *transdisciplinary research* can be conducted, particularly in health research: 'Representatives of different disciplines are encouraged to transcend their separate conceptual, theoretical, and methodological orientations in order to develop a shared approach to the research, building on a common conceptual framework. Such a framework can be used to define and analyze the research problem and develop new approaches for health care that more closely represent the historical and present-day reality in which health problems are situated' (p1351). The study team is composed of researchers and medical practitioners with a varied range of backgrounds and training. An infectious disease specialist and internist leads the core study team, which includes sociologists, anthropologists, and public health specialists who collaborate closely with a steering committee composed of clinicians trained in biomedicine and CAM, a researcher in anthroposophic medicine, public health experts and policy makers.

The project applies mixed methods through the use of a *sequential exploratory design* in order to study vaccine

## Qualitative Data Collection
*August 2017 – March 2020*

| Childhood vaccines | HPV vaccine |
|---|---|
| *August 2017 – December 2018* | *March 2019 – March 2020* |
| - Semi-structured interviews<br> • parents (N=40)<br> • providers (N=15 CAM, N=15 biomedical)<br>- Ethnographic observations of medical consultations (N=15 CAM, N=15 biomedical)<br>- Thematic and critical discourse analysis of vaccination information sources mentioned during interviews and observations | - Semi-structured interviews<br> • parents (N=15)<br> • school physicians (N=5)<br> • youth (N=25 males, N=15 females)<br> • public health officials (N=10)<br>- Family focus group discussions (N=15)<br>- Ethnographic observations of school vaccination information sessions (N=10)<br>- Thematic and critical discourse analysis of vaccination information sources mentioned during interviews and observations |

## Quantitative Data Collection
*April 2018 – March 2020*

**Questionnaires**
- Childhood vaccine questionnaire
 • Parents of children 0 to 11 years of age (N=1,350)
- HPV vaccine questionnaires
 • youth 15-26 years of age (N=722)
 • parents of youth
- Provider questionnaire
 • providers (N≈150)

**Figure 1** Study overview. CAM, complementary and alternative medicine; HPV, human papillomavirus.

hesitancy across Switzerland, meaning that we first begin with qualitative methods in order to inform the tools used for the data collection through quantitative methods.[52] An overview of the study design is shown in figure 1. We recruit participants from the three major language regions of Switzerland (German-speaking, French-speaking and Italian-speaking) in order to examine vaccination decision-making throughout the country.

The qualitative and quantitative phases of the project both involve recruitment in medical providers' offices and interviewers with medical providers. For this purpose, we established a network of participating providers via the FOPH's Swiss Sentinel Surveillance Network and through direct recruitment by our research team. Our research team and advisory board include both CAM and biomedical providers, who have helped recruit their colleagues for this project, sharing recruitment documents (ie, invitation letters and study flyers), making telephone calls and employing snowball sampling. Ongoing provider recruitment efforts have enabled us to develop a network of over 150 CAM and biomedical providers from 21 of the 26 cantons and all three language regions in Switzerland.

For the study's qualitative component, we conduct semistructured, indepth interviews with parents, youth,

and biomedical physicians and CAM providers. We also observe vaccination consultations with parents and providers and HPV information events in schools. For the study's quantitative component, we recruit participants from the practices of the network of CAM and biomedical providers across Switzerland and perform quantitative telephone interviews with parents of children 0–11 years of age and youth aged 15–26 years. We also interview the participating providers. The collection of information about vaccine hesitancy from providers and their patients allows for a comparison of their perceptions of vaccinations.

Some providers participate in both the qualitative and quantitative research components, while most participate only in the quantitative phase. Importantly, our team had already established trusting relationships with a number of CAM physicians by the time of grant submission. These contacts have been crucial in expanding our network of CAM providers in Switzerland for the quantitative study phase.

### Qualitative methods
The qualitative methods of this research allow us to gain deeper insight into parent and youth vaccination decision-making and patient–provider interactions about vaccination in general practice settings. The qualitative methods additionally further our understanding about public health gaps in vaccination implementation programmes in Switzerland by paying attention to how participants discuss the Swiss health system in relation to vaccination. Finally, the qualitative methods inform the design of the quantitative study component tools.

The transdisciplinary research team drafted interview guides and medical consultation observation guides to be used as qualitative study tools. The interview guides were written based on relevant vaccine hesitancy literature and guide researchers through semistructured interviews with participants through the use of open-ended questions. Semistructured interviews involve researchers asking the same questions systematically to all participants, but allow the option for researchers to probe further by asking additional questions about themes emerging during the interview. The medical consultation observation guides allow researchers to systematically focus on items of interest highlighted in vaccine hesitancy literature on patient–provider interactions.[28–31] Both the interview guides and the observation guides were piloted for coherence and clarity.

An important concept in qualitative research to determine sample sizes is the concept of saturation, which indicates that similar results are consistently collected during data collection and adequately address the research questions at hand. When saturation has been attained in qualitative research, researchers can reasonably expect similar results if the research process were to continue.[53 54] We expect to reach data saturation with the number of qualitative face-to-face interviews and observations mentioned

below, but the qualitative nature of this approach allows us flexibility in adjusting the amount of data to be collected.

## Qualitative interviews with biomedical and CAM providers

We conduct indepth, semistructured qualitative interviews with at least 15 CAM providers and at least 15 biomedical physicians in providers' offices. Interview questions are organised by theme, including (1) contextual information about the providers (ie, type of training, type of medicine practised, years of clinical experience and descriptions of the types of patients they treat); (2) questions about interactions with their patients and typical vaccination consultations (ie, amount of time spent discussing vaccination, communication strategies and perceptions about communication training); and (3) views and perspectives on medicine, immunity and the body (ie, information sources on vaccinations, perceptions about their advantages and disadvantages, 'natural acquired immunity' vs vaccine-acquired immunity, public health benefits compared with individual choice considerations, and vaccination rates and public health approaches in Switzerland). The interviews are audio-recorded and transcribed verbatim. If participants request transcriptions, we provide them so that they may clarify anything that they discussed.

## Ethnographic observations of vaccination consultations

We observe a total of 30 vaccination consultations (15 with CAM providers, 15 with biomedical providers) with 10 different providers (5 CAM, 5 biomedical). We ask willing providers to allow us to observe consultations during which they discuss vaccination with parents for the first time. Prior to consultation observations, researchers discuss how they are conducted, with both providers and patients having the option to not be observed. In cases of reluctance to participate, potential participants are able to ask questions about the study, and researchers provide them with written information about the study. We do not observe participants who do not wish to participate in the study. To date, providers not wishing to be observed are uncommon, and no patients have refused being observed. Providers explained these rare refusals by describing how they did not wish researchers' observations to disturb the natural flow of consultations with patients, not having enough time or not seeing enough patients with whom they discuss vaccination for the first time on a daily basis.

Researchers take ethnographic field notes about the observation in field journals in order to document what happens during the interactions. They then fill out a medical consultation observation guide, which prompts them to record information about patients and providers, the reason for the consultation, who initiated the vaccination discussion, if vaccinations were discussed in a participatory or presumptive manner, questions asked by patients, if and how patients expressed hesitancy towards vaccination, researchers' perceptions of the patients' ability to understand the vaccination discussion, the vaccination decision, the amount of time spent discussing vaccination, information source materials, and the researchers' perceptions of the mood, emotions and communication between parents and providers. After observing the medical consultations, researchers use ethnographic field notes and the observation guide as the basis to write observations into a descriptive narrative format of each observed consultation.

## Qualitative interviews with parents

We conduct indepth qualitative interviews with 30 vaccine-hesitant parents and 10 parents who vaccinate according to the FOPH vaccination schedule. Interviews are conducted in their homes or in a place of their choosing, such as a coffee shop. We purposefully sample more vaccine-hesitant parents, particularly those who consult CAM providers for their children, since an important study assumption is that vaccine hesitancy, underimmunisation by parents and their usage of CAM are codetermined. We ask parents questions thematically about (1) family composition and parental roles (ie, work, childcare, child healthcare decisions); (2) children's health, healthcare and lifestyle (ie, types of healthcare professionals parents consult for their children, health beliefs and practices, CAM use, preventive practices, and so on); (3) immunisation status of the children with the support of vaccination booklets during the interview; (4) vaccination perspectives (views on individualised vaccination calendars and views on the public health benefits or consequences of vaccination); and (5) vaccination experiences (the vaccination decision-making process, vaccination discussions with healthcare professionals and others, and perceptions of social pressure to vaccinate or not vaccinate).

## Qualitative research focus on HPV vaccination: youth and parents

There are limited published data on adolescent female views on HPV vaccine[38 55] and no data on adolescent male views on HPV vaccine in Switzerland. Therefore, we also conduct qualitative indepth interviews with approximately 40 youth (approximately 25 males and 15 females) recruited through the HPV quantitative research discussed below. Interviews take place in participants' homes or in a place of their choosing, such as a coffee shop. We purposively sample more male youth since there is no literature on male perspectives on HPV in Switzerland, and HPV vaccination recommendations are more recent for males than females. If youth provide their consent, the parents of 15 youth are interviewed separately and in family focus group discussions (FGD) together with the youth. Themes of the interviews and FGD include (1) questions relating to family composition and parental roles; (2) youth health, healthcare and lifestyle (ie, types of healthcare, professionals young adults/adolescents consult, health beliefs and practices, CAM use, preventive practices, and so on); (3) HPV immunisation status; (4) HPV perspectives; and (5) HPV vaccination experiences and the vaccination decision-making process. This last theme focuses particularly on discussions with parents, friends,

acquaintances, significant others, potential sexual partners, information sources, schools' roles in providing vaccination information, and how or if the youth ever felt any social pressure to vaccinate or not vaccinate against HPV.

### Qualitative research focus on HPV vaccination: schools and public health officials

We observe 10 school information activities about HPV vaccines in three to four cantons. We choose cantons based on convenience in terms of access and ethical clearance to conduct ethnographic observations for research purposes. Researchers attend school-based events during which the HPV vaccination is presented to youth. In classrooms, our researchers explain to students and instructors that we are there to observe and take notes on the content of the discussions as part of a study that examines how this type of information is communicated in schools. No identifying information is documented or included in any dissemination of research results. Observation notes are documented in field journals and subsequently written into a descriptive narrative format.

We purposively sample an estimated two to three public health officials from the FOPH, two to three cantonal authorities, and approximately five school physicians for expert interviews (n=10–15) in order to gain further information about how recommendations are made, discussed, planned and implemented from a public health perspective. Interviews focus on perceptions of the implementation of vaccination programmes from public health authority perspectives and are informed by specific questions from our qualitative enquiries.

### Vaccination information sources

In order to better understand how information sources affect parent, youth and provider decision-making processes, we document and analyse sources of information that they discuss during the qualitative and quantitative interviews and medical consultation observations using both thematic analysis[56] and critical discourse analysis.[57–59] Since the internet's role in vaccine-related decision-making is well documented in the literature,[14 15 60–62] we are particularly interested in parents' and youth's use of and interactions with internet sources.

### Qualitative data analysis

The following are the primary research questions guiding the conduct of the qualitative research: (1) How do parents and youth make vaccination decisions? (2) How do medical providers, both CAM and biomedical, consider what vaccination recommendations to make to their patients? (3) How do medical providers discuss vaccination with parents and youth? (4) How are vaccination implementation programmes envisaged from a public health perspective?

We use the Consolidated criteria for Reporting Qualitative research checklist[63] as guidelines for reporting items of interest in the presentation of qualitative results.

Since we work with diverse sources of qualitative data, we use the framework method[64] and grounded theory, particularly *constructivist grounded theory* as outlined by Charmaz,[65] as guiding principles for our analysis. This allows our research team to structure an analytical framework throughout qualitative data analysis with the use of *sensitising concepts* and to code data into relevant research themes while still being open to including categories and themes that emerge through qualitative data collection and analysis. Data are coded primarily by the interviewers, with coded segments being discussed with various members of the research team.

Sensitising concepts are used in qualitative research in order to guide researchers to selectively focus on emerging themes and categories relevant to the research questions[66]; for this research, sensitising concepts include patient–provider interactions, provider and patient interaction with information sources, health beliefs and practices, and interactions with and perceptions of the Swiss health system.

With the focus on patient–provider interactions in vaccination decision-making, the choice to research these questions by interviewing parents, youth and medical providers and observing vaccination consultations allows us to triangulate our qualitative data in order to account for the 'multiple realities' of medical consultations.[67] Finally, in order to fully appreciate the complexity of how people interact with health information in order to make vaccination choices, we will conduct both thematic analysis[56] and critical discourse analysis[57–59] of any sources of information mentioned as being important or determinative during vaccination discussions. A first manuscript containing initial results of the qualitative research phase has been published.[68]

### Quantitative methods

The quantitative phase of our study seeks to explore and quantify the determinants of vaccine hesitancy and underimmunisation with childhood and HPV vaccines in Switzerland. To this end, we developed questionnaires targeted at the different populations included in our study.

### The questionnaires

As childhood and HPV vaccinations involve different populations and are generally viewed differently by patients, we survey two separate populations: parents of young children and youth (and their parents, if possible). In addition, we survey providers to gain a better understanding of views on vaccination in our diverse network of participating providers and the relationship between provider and patient views on vaccination.

To fully explore vaccine hesitancy and underimmunisation in these diverse populations, we developed four separate questionnaires: (1) a childhood vaccination questionnaire to be administered to parents of young children; (2) an HPV vaccination survey for youth; (3) an HPV vaccination survey for parents; and (4) a questionnaire

for providers. As Switzerland is a multilingual country, we developed versions of the questionnaire in the three main national languages, German, French and Italian. In addition, given the large number of immigrants living in Switzerland, we developed an English version of the questionnaire for those who preferred doing the interview in English.

Parent and youth surveys include questions about basic sociodemographic factors, including household composition, work and educational background, income, migration status and language knowledge. In order to maximise comparability with other surveys and to allow us to weight samples so that results are more representative, we drew these questions from two recent representative national surveys in Switzerland: the Swiss version of the European Social Survey 2014 (ESS 2014)[69] and the 2012 Enquête Suisse de la Santé.[70]

As vaccine hesitancy is often correlated with political beliefs[71] and religious beliefs,[72] we also include a series of questions on political and religious beliefs and practices drawn from the ESS 2014. We include three questions on each theme. For religion, we ask about interviewees' religious affiliation, the frequency with which they attend religious services and their perceptions of their own religiosity. For political beliefs, we ask about interviewees' level of interest in politics, which political party they feel close to and where they place themselves of a left-right political scale.

In addition, all questionnaires include the 20-item Moral Foundations Questionnaire[73] based on the finding in a recent US study[74] showing that moral foundations are associated with vaccine hesitancy. These survey items were all already extensively published and available in our survey languages and did not require translation or validation.

We measure underimmunisation through examination of participants' vaccination records. We measure vaccine hesitancy using the 15-item Parent Attitudes about Childhood Vaccines (PACV) survey instrument, a validated measure of vaccine hesitancy and underimmunisation.[75] We use the original survey instrument for the childhood vaccination questionnaire and adapted questions to be appropriate to the target population for the parent and youth versions of the HPV questionnaire. For the provider questionnaire, we adapted the version of this instrument to measure vaccine hesitancy among childcare facility directors.[76]

Based on preliminary findings in the qualitative phase, we developed new questions around several themes, including (1) communication about vaccination with medical providers; (2) information sources consulted in vaccination decision-making; (3) perceptions about risk and control concerning possible exposure to vaccine-preventable diseases; and (4) the parent–provider relationship. For the HPV vaccination surveys, we also developed questions specific to this vaccine and its administration in Switzerland, including questions on knowledge of HPV vaccine, sources of HPV vaccination and

perceptions of school-based vaccination programmes. In addition, we ask both parents and youth a series of questions about health, including questions about health status, CAM usage and medical care usage. The provider questionnaire also asks for details about providers' practices, particularly with regard to their practice of CAM. The questions we developed that are based on preliminary findings from the qualitative phase are listed in online supplementary appendix 1.

French, German and Italian versions of the PACV15 were not available, so these questions and the new questions developed for these questionnaires were translated into French, German and Italian. Thirty-three key questions including the PACV and questions developed based on preliminary results of the qualitative portion of the study were translated using the forward and backward method. Two independent translators first translated into the target language (German, French or Italian) and then reconciled any disparities in their translations. Next, two different, independent translators who were blinded to the original questions translated them back into English.

Once developed, we pretested and piloted the childhood vaccination and HPV versions of our questionnaires in all languages. For the pretest, we recruited a convenience sample of 61 individuals to test the questionnaires in all four survey languages. We pretested each version of the questionnaire with two to seven people and adjusted based on feedback. We then piloted the adjusted questionnaires using our full recruitment procedure (see 'Recruitment and survey populations'), conducting a further 56 interviews with the three questionnaires in the four survey languages. Finally, we pretested the provider questionnaire among providers in German, French and Italian. The questionnaire was also shared with the project's advisory board and adjusted based on their feedback.

### Power analyses
#### Childhood vaccination survey

A conservative estimate is a proportion of 15% vaccine-hesitant parents in the practices of the participating biomedical and CAM providers combined (NB: 7% unvaccinated and 12% children with delayed vaccination were recorded in the 2012 FOPH survey).[77] The sample size calculation aims at identifying the number of participants needed to detect a significant association of specific individual risk factors with vaccine hesitancy, which is here considered as a binary outcome. We further assume that factors potentially associated with vaccine hesitancy are prevalent in 20% of the population and that the OR between vaccine hesitancy and a relevant risk factor is at least 2. In this case, to detect a statistically significant association with such a risk factor at the level of 0.05 with a power of 0.8, 675 participants are needed. Due to the recruitment in selected provider practices, a certain clustering of parents' characteristics is to be expected for parents recruited in the same practice, in contrast to a simple random sample. To account for such clustering, a design effect of 2 is conservatively assumed, leading to

a total sample of 1350 parents to be interviewed. This would equally allow detecting a 10% lower proportion of vaccination among parents who are vaccine-hesitant compared with a vaccination rate of 90% in the remaining population.

### HPV vaccination survey

To investigate vaccine hesitancy and associations with underimmunisation in the case of HPV vaccination, a separate sample of participants is required. Similar to the first case, we might also assume 15% vaccine-hesitant participants, and we further anticipate a 30% lower rate of vaccination among vaccine-hesitant persons than in the entire population. With a given prevalence of HPV vaccination of 53%, this would mean that among vaccine-hesitant persons only 37% would be vaccinated. With power=0.8 and alpha=0.05, a sample of 593 participants would be needed. To account for clustering, assuming a design effect of 2, the final sample would include 1186 participants. However, more likely is a higher vaccine hesitancy rate of 30%, given the relative novelty of HPV vaccines and the fact that it has been recommended in boys/adolescents only for the past 3 years. With power=0.8 and alpha=0.05, a sample of 361 participants would be needed. To account for the cluster sample, assuming a design effect of 2, the final sample would include 722 participants.

Based on the above power analyses, our study populations include 1350 parents of children aged 0–11 for the childhood vaccination study and 722 young people aged 15–21 years (for young women) or 15–26 years (for young men) for the HPV vaccination study.

As mentioned, there is regional and linguistic variation regarding *vaccination uptake* in Switzerland in the setting of different cantonal levels of school vaccination programmes[39 78] and health authority support for vaccination programmes.[79] On the other hand, we expect the prevalence of *vaccine hesitancy* and of its determinants to be relatively homogeneous across different geographical and language regions of Switzerland. Therefore, we do not specifically take geographical and language heterogeneity into account for the outcome *vaccine hesitancy*. Moreover, conditional on this assumption being correct, the statistical power to detect an association between vaccine hesitancy and the likelihood of obtaining HPV vaccination would not be sizably smaller in the presence of geographical or language heterogeneity in HPV vaccination rates than without such heterogeneity.

### Recruitment and survey populations

We recruit our participants through the offices of the network of CAM and biomedical providers. As figure 2 shows, our provider network includes more than 150 CAM and biomedical providers working in all three language

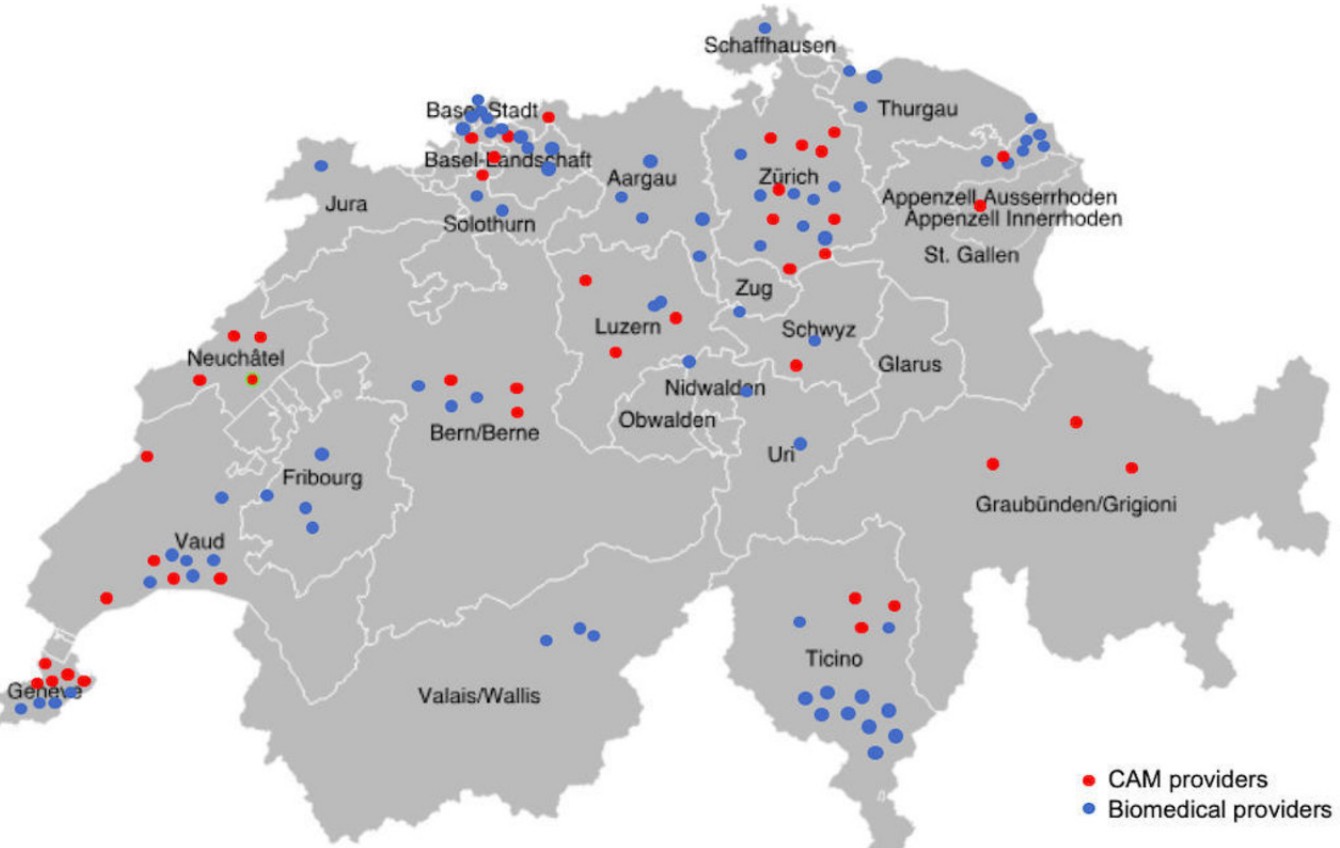

**Figure 2** Network of participating providers. CAM, complementary and alternative medicine.

regions in Switzerland and in almost every canton. Each dot represents a practice participating in the study: red dots represent CAM practitioners and blue dots biomedical providers. To ensure confidentiality, we place dots in the correct canton but do not place them at the exact provider location. Group practices are represented by a single dot.

Providers are CAM or biomedical providers who treat children and/or youth and regularly deal with questions regarding vaccinations and vaccine decisions. They include paediatricians, general practitioners, gynaecologists, CAM physicians and CAM providers without formal medical training working independently or in group practices.

We interview parents and young people by telephone. Each telephone interview lasts approximately 30–35 min. Parents are recruited from urban and rural medical practices in French-speaking, German-speaking and Italian-speaking areas of Switzerland. In addition, we interview each participating provider in order to examine the relationships between provider perspectives, patient perspectives and vaccine hesitancy. For the HPV portion of our study, we nterview one of the participating youth's parents if the youth allow us to do so.

To be eligible to participate in the study, participants fulfil the following criteria. For the childhood vaccination substudy, parents are 18 years old or older with a child aged 0–11. For the quantitative HPV portion of our study, young women are between the ages of 15 and 21 (born between 1 July 1997 and 1 July 2003) and young men are between the ages of 15 and 26 (born between 1 July 1992 and 1 July 2003). These differences are due to the timing of HPV vaccine recommendations for girls and boys. HPV vaccine reimbursement for girls 11–14 (target age) and 15–19 (catch-up age) started 1 January 2008. HPV vaccination reimbursement for males began on 1 July 2016 in Switzerland. Given the newness of this recommendation, we enrol youth males born between 1 July 1992 and 1 July 2003 in order to gather data on all males aged 15–26 who have been eligible for the HPV vaccine through Swiss vaccine programmes. All study participants (parents, youth and providers) must speak one of the main Swiss national languages (French, German or Italian) or English.

We recruit all parents and youth through providers in order to be able to link the data from the parents with their provider's data for a comparison by type of provider. We use a quota sampling technique as CAM providers are likely to see fewer patients per day. Our goal is to recruit a maximum of 20 parents of young children and a maximum of 15 youth per provider based on a rigorous sampling approach in a defined period. Recruitment targets for group practices are no more than three times the target for individual providers. Thus, targets in dual practices are 40 parents and 30 youth, and for practices with three or more providers, targets are 60 parents and 45 youth. For every practice, the total number of eligible patients in a specific time period will be established, which allows post-hoc weighting to the known total of eligible patients in this time period.

Given the diversity of providers and differences in our study subpopulations, we employ a variety of recruitment techniques. In large practices or practices with high patient volume, interviewers recruit participants directly in providers' waiting rooms. Interviewers are medical students trained in recruiting participants, obtaining informed consent, conducting quantitative interviews over the telephone and quantitative data management. For inoffice recruitment, interviewers spend 2–5 days in providers' waiting rooms, with the goal of recruiting a maximum of 20 parents and/or a maximum of 15 youth for interviews. Interviewers contact all eligible patients to present the study and ask whether they are interested in participating. For those who agree, interviewers obtain informed consent and a copy of the vaccination certificate (if possible) in waiting rooms and then arrange to conduct a telephone interview at a later time. In large group practices, interviewers have the option to work as a team to manage particularly high patient volumes.

In smaller practices or practices with lower patient volumes for the targeted groups, we ask providers to recruit participants retrospectively by reviewing their patient log from the last 2–6 months and contacting eligible patients to see if they would be interested in participating in the study. Providers transmit a detailed record of attempted contacts and take note of refusals and interested parties. We then contact those who indicate an interest by phone to present the study, obtain informed consent and a copy of the vaccination certificate (if possible), and arrange for a telephone interview. Interviewers select potential participants randomly from the lists provided until they have the required quota of participants in each group.

Interviewers determine the most appropriate recruitment technique during a first visit with the provider. As some providers may have high volumes of one population and low volumes of another (eg, paediatricians may see many young children but few youths 15 of age or over), we may employ different recruitment methods for the different subsamples.

Experience to date shows that providers generally accept our preferred recruitment practice for their type of practice. However, we work with providers to find workable solutions if they are uncomfortable with or unable to accommodate our standard recruitment procedures. Although small practices, particularly for CAM providers, are often hesitant to let interviewers recruit in waiting rooms, such practices rarely have sufficient patient volume to merit that approach. Very few large-volume providers have been unwilling to allow interviewers to recruit in waiting rooms; however, when they are, we implement the provider recruitment practice used in smaller practices. More commonly, practices do not have the capacity to review patient logs and contact patients. In that case, as approved by the local ethics commissions, interviewers can assist medical providers and their

assistance in reviewing logs and contacting patients from providers' offices.

For all recruitment techniques, we track how many people were contacted, how many agreed to participate or be contacted again, how many gave consent to interviews, and how many completed interviews. This information will allow us to assess the refusal rate, the dropout rate and the number of individuals lost to follow up.

Additionally, we conduct a quantitative telephone or face-to-face survey with all participating providers during the quantitative component of the study in order to quantitatively assess if providers' vaccine perspectives are associated with those of their patients.

## Quantitative data analysis

The following are our primary quantitative research questions:

► What are the determinants of vaccine hesitancy for childhood and HPV vaccinations in Switzerland?
► What is the relationship between vaccine hesitancy and underimmunisation?

We measure vaccine hesitancy based on the PACV score. As this instrument has not previously been used in Switzerland or in our survey languages, we first test its validity for Switzerland using Mokken Scale analysis to confirm the unidimensionality of the scale. Individuals with a PACV score of ≥50 are considered vaccine-hesitant, and individuals with a PACV score <50 are not considered vaccine-hesitant.[75] We will use multilevel logistic regression to assess the influence of different variables on vaccine hesitancy. We will explore associations between vaccine hesitancy and a number of individual-level characteristics, including age, sex, household composition, place of birth, immigration status, household income, employment status and work hours, daycare usage, highest achieved level of education, political affiliations, religion, CAM usage, trust in medical providers and satisfaction with consultations, sources of information about vaccination consulted, views on parents' role in vaccine decisions, views of importance of health, views of risks of vaccine-preventable diseases, and moral foundations. In addition, our research design allows us to explore how factors at higher levels are associated with vaccine hesitancy, including how vaccine hesitancy varies by canton and language region and by provider or by various provider characteristics (CAM vs biomedical providers), and to what extent vaccine hesitancy of the parent/youth correlates with vaccine hesitancy of the provider. Finally, although our data are not representative, we can weight results to allow us to roughly estimate the prevalence of vaccine hesitancy in Switzerland.

## Study status

This study is ongoing, with some parts of the research completed, others ongoing and others yet to begin. For the qualitative portion of the study, we have completed qualitative interviews with biomedical and CAM providers and ethnographic observations of vaccination consultations.

We have also completed qualitative interviews with parents of young children. Qualitative interviews with youth and parents focusing on HPV vaccination are ongoing. We are at the recruitment stage for our qualitative research on HPV vaccination at schools and with public health officials. Finally, we are compiling a list of vaccination information sources based on responses to questions related to this subject in the quantitative questionnaires and discussions in qualitative interviews with parents and youth. We will begin analysis once this list in complete, at the end of data collection for qualitative interviews and the questionnaires. For the quantitative portion of our study, we have completed questionnaire design and testing and the power analysis to determine sample size. We are currently recruiting and interviewing parents, youth and providers for the survey, with approximately half of interviews completed. We continue to recruit providers as well.

## DISCUSSION

By using a sequential exploratory mixed-methods study design,[52] this study will provide rich and multifaceted data on vaccination decision-making and vaccine hesitancy in Switzerland. In line with such a study design, the qualitative data informed the tools used for the quantitative data collection component. The results of both qualitative and quantitative components of our study will likely be complementary to one another and will allow us to answer different types of research questions. That said, any discrepancies or contradictory findings between the qualitative and quantitative components will merit further investigation, which we will highlight in the dissemination of study results. Additionally, given the multifaceted nature of vaccine hesitancy and health decision-making and the study's exploratory mixed-methods study design, the results from both qualitative and quantitative components will provide us with rich and detailed data for the Swiss context, where there is limited available data.

A particular strength of this study is its focus on perspectives of both medical providers and their patients. Studies have shown that providers remain a trusted source of information for parents and youth facing vaccine decisions, so a greater understanding of provider perspectives is important.[25–32] Another major strength of this study is its inclusion of the perspectives of both CAM and biomedical providers. While a growing literature explores the association between vaccine hesitancy and CAM usage,[6 12 19 20] few studies include the perspective of CAM providers and none that we know of has the extent of cooperation with CAM providers found here.

Qualitative interviews provide key insights into parents, youths, and CAM and biomedical providers' perspectives on vaccination. In addition, interviews with CAM and biomedical providers shed light onto their views on vaccination, vaccine-hesitant patients and how they interact with this population in their practices. Observations of vaccine consultations in offices provide another opportunity to get a sense of the variety of approaches providers

employ when broaching the subject of vaccination. The combination of qualitative interviews with numerous stakeholders in vaccine decision-making and planning, ethnographic observations of vaccination consultations and school information sessions about the HPV vaccination, and qualitative analysis of vaccination information sources allows for qualitative triangulation into the multifaceted phenomenon of vaccine hesitancy in Switzerland. Such a triangulation of qualitative data through multiple qualitative methods allows us to better understand the 'multiple realities' of medical consultations,[67] in addition to allowing us to explore how various actors interact with and perceive the Swiss health system in regard to vaccination.

Although data are not strictly representative of Switzerland, data from the quantitative survey gives us a sense of the overall prevalence of vaccine hesitancy in Switzerland and how this varies between language regions, cantons, and urban, suburban and rural settings. More importantly, it allows us to explore the association of vaccine hesitancy with other factors, like CAM usage, religion and spirituality, moral foundations, and political orientation. In addition, by pairing data from providers and their patients, we can see whether there is an association between providers' and patients' views on vaccines. The quantitative data also provides us with insight into the major sources of information parents and youth draw on when making vaccination decisions. Finally, the mixed-methods study design allowing for the qualitative component to inform the quantitative component also allows us to triangulate both the qualitative and quantitative data gathered throughout the duration of the study so that results from both components can be compared.

## Limitations

There are several limitations to the various parts of our study. A major limitation, which we also argue should be seen as a study strength, comes from our focus on recruiting participants through providers. This technique has major advantages when seeking to examine effects of provider characteristics on patient beliefs and behaviours, particularly for a topic such as vaccine hesitancy where trust between patients and providers has been shown to be a major determinant.[32] We also included items in both qualitative interview guides and quantitative questionnaires in order to gauge to what extent patients select physicians they trust or whose practices or attitudes align with their own, a patient practice which has been documented in vaccine hesitancy literature.[80] These questions also allow us to assess if disagreements between patients and providers have brought patients to seek care elsewhere in more provider-driven selection processes.

The quantitative component recruitment strategy indicates that populations that do not regularly see medical providers are under-represented in our data. While universal health insurance requirements attempt to ensure that most people living in Switzerland have access to healthcare, certain populations are still under-represented, including recent migrants and undocumented populations. Recruiting through providers poses further questions of selection bias, as people who see doctors more often are more likely to participate in our study. This bias means that, for example, parents of infants were more apt to be recruited than parents of older children, as infants see the doctor more regularly. Selection bias problems are particularly relevant to youth populations, as most youth are healthy and thus among the least likely to regularly seek medical care. As such, youth in our study may differ in important ways from youth in Switzerland overall. In addition, as our provider network was established through personal recruitment by members of the study team and snowball sampling, the network itself is not representative and subject to selection bias. In our view, the advantages of being able to link provider and patient data outweighed these limitations.

For qualitative interviewing, interviewer effects may affect the type and quality of information we received; interviews and observations are conducted by a male medical sociologist trained in qualitative research methods, a female medical doctor trained in qualitative research methods, a female senior medical student with qualitative research training, and a female medical anthropologist trained in qualitative research methods. That being said, having a variety of researchers with different backgrounds involved in the data collection and analytical process adds to the richness of the analysis through clarification of concepts during indepth research team discussions. Another limitation arises from the way we observed vaccine discussions. Observers watch the discussion in person and take notes, but do not audio-record or video-record discussions. The absence of such recordings limits our ability to rigorously analyse the conversations, while the physical presence of observers in the consultation may have had an effect on discussions. Furthermore, given that the data are collected in both French-speaking and German-speaking regions of Switzerland, we have a multilingual corpus which necessitates indepth discussions between study members during qualitative data analysis.

Furthermore, as noted above, the quantitative study is not based on a representative sample but instead allows us to compare results between different types of providers and their patients, and between different sociocultural regions of Switzerland. As in the qualitative component, interviewer effects may affect the type and quality of information we received. We will include data quality controls into data analysis in order to check for interviewer effects. Additionally, linguistic barriers, particularly with recent migrants to Switzerland, may prove challenging for participants to provide accurate responses to our interview questions.

## Scientific relevance

Vaccine hesitancy is a complex public health issue in high-income countries, yet there is no agreed-on measure of vaccine hesitancy. Multilingual, validated

instruments for measuring vaccine hesitancy are rare, making comparative study difficult. This study develops and test translations of the 15-item PACV in three new languages and provides valuable assessments of vaccine hesitancy in French-speaking, German-speaking and Italian-speaking contexts. Furthermore, our study design includes vaccination certificates in order to statistically examine the relationship between vaccine hesitancy and underimmunisation.

Results from the study make a contribution to the literature on the relationship between CAM usage and vaccine hesitancy. The combination of rich qualitative data including both biomedical and CAM perspectives and provider-linked data from patients who see CAM practitioners allows for a more indepth examination of this relationship than found in the literature to date.

A combination of the data collected through the quantitative and qualitative methods and their analysis will form the necessary background for the design, submission and implementation of an appropriate randomised control trial intervention designed at improving vaccine communication and counselling among physicians, parents and adolescents in Switzerland.

## Patient and public involvement

Study participants, both patients and medical providers, are involved in the study design throughout the duration of the data collection. Study tools are piloted and tested by the research team who requests patient and provider input during this process. Participants are given the opportunity to indicate any additional comments throughout data collection so that we may take their priorities, experiences and preferences into account. Patients (parents and adolescents) and providers assist us in additional recruitment efforts as described in the Methods and design section. All participants were informed of the amount of time their participation would entail prior to data collection. We systematically request participants' wishes to receive the results of the study by including an item on the consent form allowing them to provide us with their contact information. We will send relevant published results to study participants and invite them to public forums where we discuss our study results.

## ETHICS AND DISSEMINATION

As approved by the ethics committee, all questionnaire items include 'do not wish to answer' categories, given the personal and sensitive nature of certain questions, in particular those regarding religion and political views. In accordance with the Human Research Act (HRA), adolescents (defined as individuals 14 years of age or older) who are capable of judgement are able to provide informed consent in writing and do not need to provide informed consent in writing from a legal representative 'if the research project entails more than minimal risks and burdens' (HRA, Art 23, Par 1). Ethikkommission Nordwest- und Zentralschweiz has approved of this study

as entailing less than minimal risks and burdens and thus does not require informed consent in writing from a legal representative for adolescents. Informed consent from minor participants (all aged 15 years and older for this study) is obtained directly from the participants and not their legal representatives.

Study results will be published in peer-reviewed academic journals and disseminated to healthcare professionals, researchers and the public via academic conferences and public presentations. The data sets used and/or analysed during the current study are available from the corresponding author on reasonable request.

**Author affiliations**
[1]Swiss Tropical and Public Health Institute, Basel, Switzerland
[2]University of Basel, Basel, Switzerland
[3]Department of Pediatrics, HFR Fribourg Cantonal Hospital, Fribourg, Switzerland
[4]University Department of Medicine, Kantonsspital Baselland, University of Basel, Bruderholz, Switzerland
[5]Klinik Arlesheim, Arlesheim, Switzerland
[6]Institute of Sociological Research, University of Geneva, Geneva, Switzerland

**Presented at**
Some of the study's methods and preliminary findings have previously been presented at the following conferences: (1) Notter J, Deml M, Huber B, Krüerke D, Jafflin K, Zeller A, Mäusezahl M, Berger C, Merten S, Burton-Jeangros C, Pfeiffer C, Tarr PE. Poster presentation at SSI (Swiss Society for Infectious Diseases) Joint Annual Meeting in Interlaken, 14 September 2018: 'Complementary and alternative medical (CAM) providers' patient-centered, individualized approaches to vaccination in Switzerland'; (2) Tarr P. Oral presentation at Global challenges in vaccine acceptance science and programs, 24–26 September 2018 - Les Pensières Centre for Global Health, Veyrier-du-Lac, France: 'Approaches of complementary and biomedical providers to vaccinations in Switzerland'; (3) Deml M. Oral presentation at Swiss Public Health Conference in Neuchâtel, 8 November 2018: 'Vaccination Policy, Vaccine Uptake, and Vaccine Hesitancy in Switzerland'.

**Acknowledgements** The authors would like to thank Christoph Berger (University Children's Hospital Zurich), Andreas Zeller (University Institute of Primary Care Medicine Basel), Mirjam Mäusezahl (Epidemiology Section, Federal Office of Public Health, Bern), Julia Notter (Medizinische Universitätsklinik, Kantonsspital Baselland, University of Basel, Switzerland) and Christian Schindler (Senior Statistician, Swiss Tropical and Public Health Institute, University of Basel, Switzerland) for their important contributions to protocol development and for many helpful discussions.

**Contributors** MD codrafted the manuscript and focused on the qualitative components. KJ codrafted the manuscript and focused on the quantitative components. SM initiated the drafting of the manuscript and provided valuable feedback during its writing. BH assisted in establishing a network of complementary and alternative medical (CAM) practitioners and provided expertise in both paediatrics and CAM in the Swiss context. AB participated in developing the methods for the qualitative research specific to HPV. EF, VM, JS, PK and RC coordinated quantitative data collection efforts and were instrumental in the preparation of the study's quantitative tools and data management. DK participated regularly in study advisory board meetings and provided insight into CAM perspectives and feedback about the study design. CP took a lead role in establishing the study's qualitative methodologies. CB-J provided regular study supervision and inputs about the qualitative results. PT is the principal investigator, directed the funding request and supervised the conduct of the study in its entirety. He provided infectious disease and general medical expertise, and oversaw study conception, design, data collection, analysis and interpretation. All authors read, contributed to and approved the final manuscript.

**Funding** The study is funded in its entirety via the Swiss National Science Foundation for 4 years (2017–2021), Grant Number 407440_167398, in the setting of National Research Programme (NRP) 74. No funding was obtained from vaccine manufacturers or the Federal Office of Public Health.

**Map disclaimer** The depiction of boundaries on the map(s) in this article do not imply the expression of any opinion whatsoever on the part of BMJ (or any member of its group) concerning the legal status of any country, territory, jurisdiction or

area or of its authorities. The map(s) are provided without any warranty of any kind, either express or implied.

**Competing interests** None declared.

**Patient consent for publication** Not required.

**Ethics approval** The study is conducted in compliance with the study protocol, the Swiss Federal Act on Research Involving Human Beings (Human Research Act) and the Declaration of Helsinki. The study was approved by the local ethics committee (Ethikkommission Nordwest- und Zentralschweiz, EKNZ; project ID number 2017–00725), and ethics approval covers all study participants, including adolescent and adult patients, as well as biomedical and CAM providers. All study participants receive information about the study and provide written informed consent.

**Provenance and peer review** Not commissioned; externally peer reviewed.

**Data availability statement** Data are available upon reasonable request.

**Open access** This is an open access article distributed in accordance with the Creative Commons Attribution 4.0 Unported (CC BY 4.0) license, which permits others to copy, redistribute, remix, transform and build upon this work for any purpose, provided the original work is properly cited, a link to the licence is given, and indication of whether changes were made. See: https://creativecommons.org/licenses/by/4.0/.

**ORCID iD**

Michael J. Deml http://orcid.org/0000-0003-2224-8173

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
