## [Reviewer comments · BMJ Open]

ARTICLE DETAILS

TITLE (PROVISIONAL)	Determinants of Vaccine Hesitancy in Switzerland - Study Protocol of a Mixed Methods National Research Program
AUTHORS	Deml, Michael; Jafflin, Kristen; Merten, Sonja; Huber, Benedikt; Buhl, Andrea; Frau, Eleonora; Mettraux, Valérie; Sonderegger, Joanna; Kliem, Paulina; Cattalani, Rachele; Krüerke, Daniel; Pfeiffer, Constanze; Burton-Jeangros, Claudine; Tarr, Philip

VERSION 1 – REVIEW

REVIEWER	Verger pierre Observatoire régional de la santé PACA Marseille/France
REVIEW RETURNED	11-Jul-2019

GENERAL COMMENTS	Determinants of Vaccine Hesitancy in Switzerland – Study protocol of a Mixed Methods National Research Program This a very relevant, rich (and also complex) ongoing research program which should allow to better understand the determinants of vaccine hesitancy (VH) in this country, but, above all, the role of patient-provider interactions in the dynamic of VH. Apart the mixed method (qualitative and then quantitative) method, which is a strength of this program, the fact that it will allow a focus on complementary and alternative medicine providers and also, the comparison of patients attitudes and behaviors to those of their providers, are very interesting and original aspects of the protocol. The article is very well written, in a detailed way, which allows apprehending the full program in its variety and complexity. Below are listed some points that need clarification however. Ethical aspects: the program complies with the Swiss Federal act on Research Involving Human Beings, and has been approved by a local ethics committee. But the reviewer is surprised that information will be collected with questionnaires on political views, and religion: this is very personal and sensitive personal information, which researchers are not always allowed to collect, in certain countries. More information should be provided on this point in the manuscript, including, in the method section, on which questions will be used. Objectives. Given the rich and ambitious research program presented in this article, the objectives section appears written in a somewhat general way. The reviewer wonders whether the authors could have proposed some complementary specific objectives which would also shed light on the analysis section. For example, as vaccine hesitancy will be studied in all groups (including providers, if the reviewer has well understood (lines 471 to 477)), it would be interesting to assess specifically whether VH in patients is associated with VH in providers. Moreover, do the
---

	authors aim at quantifying the impact of different potential determinants on patients VH at various levels: patients, providers, territories (which various organizational, language and cultural characteristics)? Design. The research team applies a sequential mixed method, first beginning with qualitative methods in order to inform the collection tools to be used in the quantitative surveys. As the program is ongoing, it would be useful to add some information on the extent to which the first qualitative phase helped designing the quantitative questionnaires and on which aspects, more precisely than evoked in the manuscript. There is also a second question: did the research team consider the possibility of a third phase, qualitative, to better understand some of the results of the quantitative phase? Selection procedure of providers: the way the providers network was established would merit some more information. Feasibility: questions about the feasibility of the program arise especially regarding: 1) the ethnographic phase: how was/is it accepted how and did the research team managed to overcome the probable reluctances of both patients and providers? 2) the quantitative phase: to which extent was it difficult to find health providers accepting the presence of interviewers in their waiting room? What about providers' acceptance, in small practices, to proceed to recruitment of patients, record and transmit detailed follow-up of attempted contact? To which extent accepting these procedures may have led to selection bias? How did the authors anticipate (and manage) secondary refusals (if any) of providers, and of patients (to answer a telephone questionnaire)? Statistical power of the quantitative surveys: in the estimation of the number of subjects in the two patients surveys, did the authors take into account the clustering effect potentially associated with language and territory? The paragraph concerning the HPV vaccination survey is unclear: first, assuming the same rate of vaccine hesitancy as for the childhood vaccination survey should be further discussed: there are reasons (including its novelty) to consider that vaccine hesitancy regarding the HPV vaccine can be different from that regarding childhood vaccines. Then, it is unclear why the authors present two estimations, and between them, chose to keep the least conservative (the smallest one, which will not provide sufficient power, if the hypothesis of lower patient VH in this survey is not true). Analysis: The richness of the program lies on the mixed –methods. The authors write in their protocol that they will use triangulation to address both the variety of points of view gathered in this program within the qualitative phase, and also the results of the qualitative and quantitative phases. This is somewhat general and further information or clarification of the specific methods/approach that will be used to ensure the “mixed analysis” of the data would be welcome. Regarding the quantitative surveys, did the authors consider weighting their samples of patients, as there are not representative? This would be preferable if the authors aim at providing an overall prevalence of VH (lines 667-68). Given the hierarchical structure of data (territory, practice, patients), the use of multiple logistic regression is questionable; multilevel analyses may be more appropriate. Limitations. The authors write (line 684) that “recruiting patients through providers has major advantages when seeking to examine effects of provider characteristics on patient’s beliefs and
--	---

	behaviors.” The reviewer agree with this, but this raises the question of endogeneity however. This means that patients, in a system where they are free to choose their health provider, may engage resources to choose the provider who best fits their expectations (see Peretti-Watel, Sociological health and illness, 2019). On the other side, some providers (not the majority, fortunately), will also apply a certain form of patient selection, especially in the field of vaccination: some providers (especially pediatricians) will refuse to take care of patients refusing vaccination or some vaccines). These endogeneity mechanisms will act as confounding factors of the correlation between patients and providers attitudes toward vaccination. How will the author address this limitation?
--	---

REVIEWER	Heidi Larson London School of Hygiene& Tropical Medicine
REVIEW RETURNED	11-Jul-2019

GENERAL COMMENTS	Overall, I think this is an important study and brings a rare multi-disciplinary, mixed-methods approach to the complex issue of vaccine hesitancy. The limitations are clear and well articulated, but given the multiple dimensions of the research and variety of different study participants, the risk is how will this all be analysed and brought together in a cohesive, meaningful outcome? It would also be helpful to summarize all the different strands of research methods/study populations/time frame, etc. into one table as it currently reads more like a long list of different sub-studies without a clearly articulated overview.
---

VERSION 1 – AUTHOR RESPONSE

Reviewers' Comments to Author:

Reviewer: 1

Reviewer Name: Verger, Pierre

Institution and Country: Observatoire régional de la santé PACA Marseille/France

Please state any competing interests or state 'None declared': None

This a very relevant, rich (and also complex) ongoing research program which should allow to better understand the determinants of vaccine hesitancy (VH) in this country, but, above all, the role of patient-provider interactions in the dynamic of VH. Apart the mixed method (qualitative and then quantitative) method, which is a strength of this program, the fact that it will allow a focus on complementary and alternative medicine providers and also, the comparison of patients attitudes and behaviors to those of their providers, are very interesting and original aspects of the protocol. The article is very well written, in a detailed way, which allows apprehending the full program in its variety and complexity. Below are listed some points that need clarification however.

Ethical aspects: the program complies with the Swiss Federal act on Research Involving Human Beings, and has been approved by a local ethics committee. But the reviewer is surprised that information will be collected with questionnaires on political views, and religion: this is very personal and sensitive personal information, which researchers are not always allowed to collect, in certain countries. More information should be provided on this point in the manuscript, including, in the method section, on which questions will be used.

Response: Please find a detailed overview of the source and nature of these questions in the methods section on pages 20-21, lines 537-561. Importantly, and as approved by the ethics committee, all questionnaire items include “do not wish to answer” categories, given the personal and sensitive nature of certain questions, in particular those regarding religion and political views. This has been added to the revised manuscript on page 36, lines 1003-1006.

Objectives. Given the rich and ambitious research program presented in this article, the objectives section appears written in a somewhat general way. The reviewer wonders whether the authors could have proposed some complementary specific objectives which would also shed light on the analysis section. For example, as vaccine hesitancy will be studied in all groups (including providers, if the reviewer has well understood (lines 471 to 477)), it would be interesting to assess specifically whether VH in patients is associated with VH in providers. Moreover, do the authors aim at quantifying the impact of different potential determinants on patients VH at various levels: patients, providers, territories (which various organizational, language and cultural characteristics)?

Response: We have clarified these points in the section on quantitative data analysis on pages 28-29, lines 797-810. The various items we postulate as correlates to vaccine hesitancy and under-immunization are itemized in objective 1 (page 9, lines 257-260).

Design. The research team applies a sequential mixed method, first beginning with qualitative methods in order to inform the collection tools to be used in the quantitative surveys. As the program is ongoing, it would be useful to add some information on the extent to which the first qualitative phase helped designing the quantitative questionnaires and on which aspects, more precisely than evoked in the manuscript.

Response: We have reorganized and expanded our discussion of questionnaire design so as to clarify how findings from the qualitative phase informed questionnaire development. Please find the discussion of this specific point on page 21, lines 570-575.

We have also included the additional questions as a separate file, Appendix 1, for supplementary materials for the study protocol.

There is also a second question: did the research team consider the possibility of a third phase, qualitative, to better understand some of the results of the quantitative phase?

Response: This is an important point and clearly a possibility we are considering. However, this discussion is in very early stages and it would be too speculative to substantially state anything about this in the current manuscript.

Selection procedure of providers: the way the providers network was established would merit some more information.

Response: We have reorganized and expanded our discussion of this point in the introduction to our methods section on page 11, lines 294-303.

Feasibility: questions about the feasibility of the program arise especially regarding: 1) the ethnographic phase: how was/is it accepted how and did the research team managed to overcome the probable reluctances of both patients and providers?

Response: We have clarified this on page 14, lines 376-386.

2) the quantitative phase: to which extent was it difficult to find health providers accepting the presence of interviewers in their waiting room? What about providers' acceptance, in small practices,

to proceed to recruitment of patients, record and transmit detailed follow-up of attempted contact? To which extent accepting these procedures may have led to selection bias? How did the authors anticipate (and manage) secondary refusals (if any) of providers, and of patients (to answer a telephone questionnaire)?

Response: We have added a paragraph discussing our experience with this to date in the Recruitment and survey populations section on page 27-28, lines 769-785. As discussed there, we have not had much difficulty finding large practices willing to allow us to recruit in waiting rooms. We have had more difficulty with small practices' capacity to carry out retrospective recruitment.

We have also added a discussion of how we track contacts, refusals and people lost-to-follow-up, which can be found on page 28, lines 781-785. We have also added a discussion of the implications our provider recruitment methods have for selection bias to the limitations section on p. 33, lines 926-929.

Statistical power of the quantitative surveys: in the estimation of the number of subjects in the two patients surveys, did the authors take into account the clustering effect potentially associated with language and territory?

Response: We have clarified these points on page 24, lines 677-687.

The paragraph concerning the HPV vaccination survey is unclear: first, assuming the same rate of vaccine hesitancy as for the childhood vaccination survey should be further discussed: there are reasons (including its novelty) to consider that vaccine hesitancy regarding the HPV vaccine can be different from that regarding childhood vaccines. Then, it is unclear why the authors present two estimations, and between them, chose to keep the least conservative (the smallest one, which will not provide sufficient power, if the hypothesis of lower patient VH in this survey is not true).

Response: There was a typo in this paragraph. Because the recommendation to vaccinate also boys/young men is relatively new, it is reasonable to expect a higher (not, as we stated, lower) than 15% HPV vaccine hesitancy rate. This has been corrected in the revised manuscript, on page 24, lines 668.

Analysis: The richness of the program lies on the mixed –methods. The authors write in their protocol that they will use triangulation to address both the variety of points of view gathered in this program within the qualitative phase, and also the results of the qualitative and quantitative phases. This is somewhat general and further information or clarification of the specific methods/approach that will be used to ensure the “mixed analysis” of the data would be welcome.

Response: We have clarified this point in the Discussion section, pages 30-31, lines 853-863.

Regarding the quantitative surveys, did the authors consider weighting their samples of patients, as there are not representative?

This would be preferable if the authors aim at providing an overall prevalence of VH (lines 667-68).

Response: We have added a discussion of this point in our discussion of questionnaire design on page 20, lines 537-541. In short, yes, we purposefully borrowed socio-demographic questions from recent nationally representative surveys to ensure that we would be able to do this.

We have also added a short discussion of this in the quantitative analysis section of p. 29, lines 810-817.

Given the hierarchical structure of data (territory, practice, patients), the use of multiple logistic regression is questionable; multilevel analyses may be more appropriate.

Response: We agree that multilevel logistic regression is more appropriate here, and we have revised to make it clear that this is the method we will use for analysis on p. 29, line 802.

Limitations. The authors write (line 684) that “recruiting patients through providers has major advantages when seeking to examine effects of provider characteristics on patient’s beliefs and behaviors.” The reviewer agree with this, but this raises the question of endogeneity however. This means that patients, in a system where they are free to choose their health provider, may engage resources to choose the provider who best fits their expectations (see Peretti-Watel, Sociological health and illness, 2019). On the other side, some providers (not the majority, fortunately), will also apply a certain form of patient selection, especially in the field of vaccination: some providers (especially pediatricians) will refuse to take care of patients refusing vaccination or some vaccines). These endogeneity mechanisms will act as confounding factors of the correlation between patients and providers attitudes toward vaccination. How will the author address this limitation?

Response: In this cross-sectional study, we cannot assess any causal link between provider vaccine attitudes and patient vaccine attitudes. We agree that any correlation between the two is driven by the sorts of selection mechanisms that you mention. We have clarified how patients’ physician selection practices had been included as items for the data collection tools, p. 32-33, lines 908-914. Questions in the quantitative questionnaire also allow us to assess the extent to which provider-driven selection processes may be at work. We have included these questions in Appendix 1 and included this item with the questions developed based upon preliminary qualitative findings.

Reviewer: 2

Reviewer Name: Heidi Larson

Institution and Country: London School of Hygiene& Tropical Medicine

Please state any competing interests or state ‘None declared’: none

Overall, I think this is an important study and brings a rare multi-disciplinary, mixed-methods approach to the complex issue of vaccine hesitancy. The limitations are clear and well articulated, but given the multiple dimensions of the research and variety of different study participants, the risk is how will this all be analysed and brought together in a cohesive, meaningful outcome?

Response: We agree. Given the multifaceted and complex nature of vaccine hesitancy in the Swiss context, we cannot reasonably argue that there will be one single, meaningful outcome. Rather, this research will serve as a fruitful starting point for research into vaccine hesitancy and under-immunization in the Swiss context, where such data is currently limited and disparate. We have further clarified this in the discussion section, p. 30-31, lines 853-863.

It would also be helpful to summarize all the different strands of research methods/study populations/time frame, etc. into one table as it currently reads more like a long list of different sub-studies without a clearly articulated overview.

Response: Agreed. We have included a table entitled Figure 1: Study Overview (mentioned p. 10, line 286) which will be included in the body of the text of the manuscript.

VERSION 2 – REVIEW

REVIEWER	Pierre Verger Southeastern Health Regional Observatory, Marseilles, France
REVIEW RETURNED	10-Sep-2019
GENERAL COMMENTS	I have just one comment left: Appendix 1 is far too long and would benefit simplification and reduction (by choosing only one language, for example).